# OMG: Orthogonal Method of Grouping With Application of K-Shot Learning

**Haoqi Fan**      **Yu Zhang**      **Kris M. Kitani**
The Robotics Institute, School of Computer Science
Carnegie Mellon University
Pittsburgh, P.A., 15213

## Abstract

Training a classifier with only a few examples remains a significant barrier when using neural networks with a large number of parameters. Though various specialized network architectures have been proposed for these k-shot learning tasks to avoid overfitting, a question remains: is there a generalizable framework for the k-shot learning problem that can leverage existing deep models as well as avoid model overfitting? In this paper, we proposed a generalizable k-shot learning framework that can be used on any pre-trained network, by grouping network parameters to produce a low-dimensional representation of the parameter space. The grouping of the parameters is based on an orthogonal decomposition of the parameter space. To avoid overfitting, groups of parameters will be updated together during the k-shot training process. Furthermore, this framework can be integrated with any existing popular deep neural networks such as VGG, GoogleNet, ResNet, without any changes in the original network structure or any sacrifices in performance. We evaluate our framework on a wide range of k-shot learning tasks and show state-of-the-art performance.

## 1 Introduction

As many deep learning network architectures have been proposed, several key models have emerged as default models for many image classification tasks. In particular, deep neural network architectures such as the VGG network by Karen & Zisserman (2014), Inception by Christian et al. (2015) and ResNets by Kaiming et al. (2015), have proven their superb performance on image classification when using datasets such as ImageNet (Alex et al. (2012)) and MS COCO (Lin et al. (2014)). With enough data and fine tuning, these 'go to' models have been shown to be successful for many visual classification tasks.

However, there is a problem when one does not have access to a large labeled training dataset to fine-tune these models. This task of training a classifier using only a small $k$ number of examples is often referred to as $k$-shot learning and has problems when dealing with high capacity models such as deep convolutional neural networks. The problem of training with only a small number of training examples is that it often leads to overfitting, where the model essentially memories those data points without gaining the ability to generalize to new instances.

Current state-of-the-art methods (Oriol et al. (2016) Adam et al. (2016)) apply deep learning techniques by using specialized network architectures for $k$-shot learning to avoid overfitting. While this is a reasonable strategy for proposing diverse kinds of new frameworks for $k$-shot learning tasks, it is hard for different $k$-shot learning methods to borrow structures from each other since they are all highly customized networks.

We propose a method called the *Orthogonal Method of Grouping* (OMG) to facilitate a better $k$-shot learning process. OMG ensures that similar (near duplicate) features in the classifier will be grouped and modified together during the training process. This process of grouping features naturally induces dimension reduction of the parameter space and imposes a form of subspace regularization during training. We implement OMG by adding a new loss layer that essentially clusters (groups) parameters by slighting perturbing them according to an orthogonality constraint. This para-loss layer only augments the network and does not require any changes to the original architecture. Once the

feature has been grouped, the network can be used to learn from only a few examples and parameter updates are propagated to parameter groups instead of individual parameter updates.

Our contribution is threefold: (1) we proposed a general $k$-shot learning approach which does not rely on any task-specific prior knowledge; (2) our approach can be added to any network without changing the original network structure; and (3) the proposed method provides an effective technique for decomposing the parameter space for high capacity classifiers.

## 2   RELATED WORKS

One-shot learning is an interesting topic first presented in Li et al. (2006). The key idea of one-shot learning is to make a prediction on a test instance by only observing a few examples of one-shot classes before. Li et al. (2006) solved this problem by adopting Variational Bayesian approach where object categories are represented by probabilistic models. More recently, researchers revisited one-shot learning with highly customized models: M. et al. (2011) address one-shot learning for character recognition with a method called Hierarchical Bayesian Program Learning (HBPL). It modeled the process of drawing characters generatively to decompose the image into small pieces. The goal of HBPL is to determine a structural explanation for observed pixels. However, inference under HBPL is difficult since the joint parameter space is very large, which leads to an intractable integration problem. Gregory (2015) also presented a strategy for performing one-shot classification by learning deep convolutional siamese neural network for verification. Oriol et al. (2016) proposed a Matching Nets utilizing external memory with attention kernel. Adam et al. (2016) proposed the memory-augmented neural networks with an external content based memory. These works achieve state-of-the-art performance on specific datasets. However, one important issue would be, works mentioned above proposed highly customized networks, whose structures is hard to be borrowed from other. Conversely, OMG is a general one-shot learning model can fit into any existing networks. So its structure could borrow by any other works.

Domain Adaptation is another related topic to our work. It aims at learning from a source data distribution a well-performing model on a different (but related) target data distribution. Many successful works as Boqing et al. (2012) and Basura et al. (2013) seek an embedding of transformation from the source to target point that minimizes domain shift. Daume III (Daum III (2009)) is another simple feature replication method that augments feature vectors with a source component, a target component, and a shared component. Then an SVM is trained on the augmented source and target data. These methods are proven to be effective for many tasks, but none of these methods above could feed into an end to end learning framework. One end to end learning framework would be the supervised adaptation method proposed by Judy et al. (2013a). It trains different deep networks for source and target domain and concatenating the high-level features as final embedding. However, the limitation is - the high-level features trained by one domain can not borrow knowledge from the other domain. In contrast, our method is an end-to-end method learning on target data without training additional networks. So the OMG can contain knowledge from both source and target domain, which is to say, the source and target domain could borrow knowledge from each other.

Our OMG model could decompose existing feature to a compact feature space, which is closely related to dimension reduction. Many cookbook works such as Aapo (1999) Jolliffe. (1986) Najim et al. (2011) have been studied extensively in the past. However, these works can't integrate into any current end-to-end deep learning frameworks. As far as we know, there is limited work having visited the topic of end-to-end dimension reduction. E. & Salakhutdinov (2006) used auto-encoder as a dimension reduction method. Stacked restricted Boltzmann machine (RBM) is proposed to embed images to lower dimension. However, this work is also hard to be integrated into existing network structures or utilize pre-trained models. The reason is that it has a different architecture than CNN networks, and it requires to train from scratch. Our OMG is an end-to-end approach that's able to feed into any architectures smoothly for both training and finetuning. Furthermore, OMG could reduce the output to an arbitrary dimension without changing the architecture and sacrifice the performance.

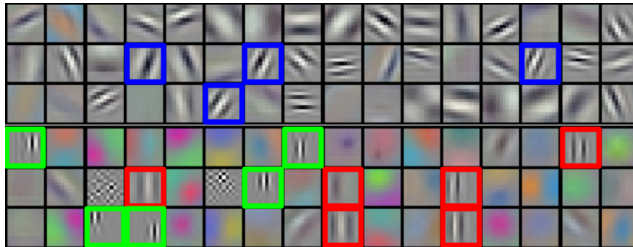

Figure 1: One illustration of the parameter basis in VGG net. Visualization of the first convolutional layer from VGG Karen & Zisserman (2014) network pre-trained on ImageNet (Alex et al. (2012)). Filters (parameter basis) with the same color of the bounding box are correlated. These filters in the same color are functionally similar, which mean they will have similar activation given same input.

## 3 APPROACH

We observe the fact that the parameters of each layer of the deep network are always correlated. This can be best illustrated by Fig. 1. The correlated parameters will result in correlated outputs with a lower capacity. When learning new classifier on top of these correlated outputs, it is easier to get an instance specific classifier if only small amount of data is seen. In most of the cases, an instance specific classifier is not what we always want. If the correlation of output is removed, which output is becoming orthogonal, then a better classifier will be fetched given a few data. So we proposed an Orthogonal Method of Grouping (OMG) to remove the correlation of outputs by decomposing the correlated the parameters to orthogonal parameters.

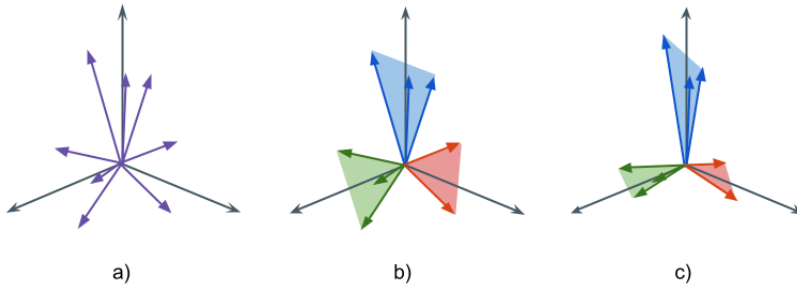

Figure 2: The initial parameters as shown in (a), where parameters are correlated. (b) is an illustration of our grouping algorithm, the algorithm assigns each different parameters to a corresponding group with a one to one mapping. (c) illustrates our algorithm cast a constraint to force parameters from each group orthogonal to those from other groups.

### 3.1 ORTHOGONAL METHOD OF GROUPING

OMG is a two-step method which can be best illustrated by looking at Figure 2. In Figure 2 (a), each purple vector represent a parameter.

In the first step of OMG (Figure 2 (b)), it finds correlated parameters and groups them in the same subspace. Since correlated parameters will result in correlated outputs, we can find the relation between parameters by analysis the output (e.g., the relation between convolutional kernels could be found by analysis the activation of the convolutional layers). In the second step of OMG (Figure 2 (c)), each parameter is slightly perturbed such that the orthogonality between each grouped subspace is maximized. The groups are represented as an Orthogonal Group Mapping that each parameter is mapped to the corresponding group. The mapping is learned by optimizing the orthogonal constraint on both $\theta_w$ and $\theta_{map}$, where $\theta_w$ is the parameters of the neural networks, and $\theta_{map}$ is the parameters of the Orthogonal Group Mapping.

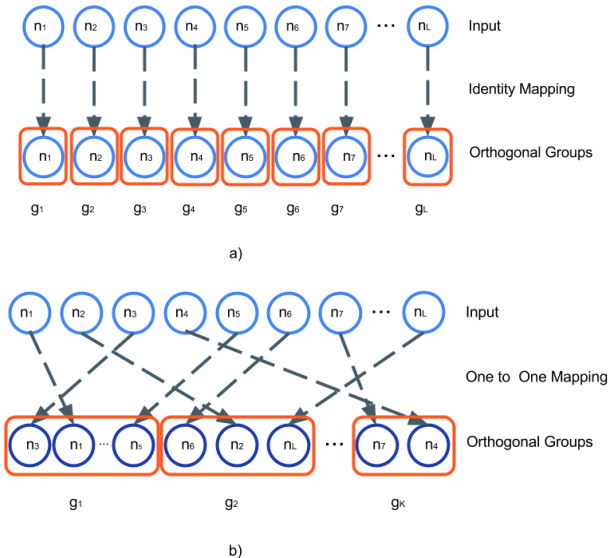

Figure 3: This figure illustrates the framework of the Orthogonal Method of Grouping. $n_1...n_M$ represent the $M$ different neural units (basis vectors). The dotted arrows represent the one to one mapping from each neural unit to their corresponding orthogonal groups. Each orthogonal group $g_i$ is represented as a red square. In a) it illustrates a special case of Orthogonal Method of Grouping with identity mapping. This special case can represent the connection between any normal layers. This means every normal layer are a special case of OMG. (b) is a normal case of OMG learning the one to one mapping from neural units to their corresponding orthogonal groups.

The Orthogonal Group Mapping is firstly introduced in Sec 3.1.1, followed with loss function and orthogonal constraint in Sec 3.1.2. Then the optimizing algorithm is introduced in Sec 3.1.3. Finally, the k-shot learning method learned on orthogonal grouped parameters is introduced in 3.2

### 3.1.1 ORTHOGONAL GROUP MAPPING

Orthogonal Group Mapping (OGM) maps neural units to the corresponding groups by a mapping. Let neural units in a layer of the network be denoted as a set $\mathbf{n} = \{n_1, n_2, ..., n_L\}$, where $L$ is the number of neural unit in that layer. (For example, a filter of a fully convolutional layer can represent as a unit.) The Orthogonal Group Mapping could represent as orthogonal group sets $\mathbf{g}$, where $g_k$ is the $k$th orthogonal group in $\mathbf{g}$. Each orthogonal group $g_i$ contains the corresponding units $g_i = \{n_j, ..., n_l\}$. Since the OGM is a map that, one unit is only mapped to one single group.

### 3.1.2 PARA-LOSS

We cast constraints to force orthogonal groups orthogonal to each other. The constraint is achieve by a loss function with two terms: intra-class loss $L_{intra}$ and inter-class class $L_{inter}$. $L_{intra}$ minimize the divergence in each group, and $L_{inter}$ force the basis in each orthogonal group orthogonal to each other.

When a mini-batch of input data of size $B$ is propagated through the network, the output (activation) of units $\mathbf{n}$ over the mini-batch can be denoted as a matrix $A \in R^{B \times L}$, where each element $A_l \in R^{1 \times L}$ denotes the output of the $l$-th unit over the entire mini-batch. Intuitively, if the outputs $A_i$ and $A_j$ of two units $n_i, n_j$ are similar, we would like to let the two units belong to the same orthogonal group $g_k$. Conversely if the outputs $A_i$ and $A_j$ of two units $n_i, n_j$ are different, then they should belong to different orthogonal groups.

We define the intra-group loss by the sum of squared distances between each orthogonal groups:

$$L_{intra} = \sum_k \sum_{i,j \in g_k} ||A_i - A_j||_2$$

The time complexity is $\Theta(L_{intra}) = K \times k_{\max} \times k_{\max}$, where $k_{\max} = \max_i k_i$. This could be efficiently computed when $k_{\max}$ is small, but we still want to reduce $\Theta(L_{intra})$ since the computational cost of this loss can be significant when there are many units in a single layer. We can use a lower bound to approximate the distance:

$$\tilde{L}_{intra} = \sum_k \sum_{i \in g_k} (A_i - A_{anchor})^2$$

where $anchor$ is an index randomly selected from the $k$-th orthogonal group. This approximation reduces the time complexity to $\Theta(\tilde{L}_{intra}) = K \times k_{\max} \approx L$, which is linear.

In addition to quantifying the intra-group similarity (tightness of each cluster), we also want to measure the separation between each group. We can define an inter group loss in terms of an orthogonality measure:

$$L_{inter} = \sum_{i,j} ||M_i^\top M_j||_F^2$$

Where matrix $M_i \in R^{B \times |g_i|}$ represents the output of all the units in orthogonal group $g_i$ over the mini-batch, where $|g_i|$ is the number of units in the orthogonal group $g_i$. That is to say, $M_i = [A_j, ..., A_k]^T$, where $n_j, n_k \in g_i$. The $|| \cdot ||_F^2$ is the squared Frobenius norm. This term is minimized when feature vectors are exactly orthogonal.

The entire para-loss function is denoted as:

$$L = \alpha \sum_k \sum_{i \in g_k} (A_i - A_{anchor})^2 + \beta \sum_{i,j} ||M_i^\top M_j||_F^2$$

It is easy to see the $L$ does not contain any term of ground truth label. So the OMG is an unsupervised method without requiring ground truth. However, this method could work smoothly with other losses. For example, OMG could train with loss of softmax $L_{softmax}$ simultaneously for a supervise learning task.

### 3.1.3 OPTIMIZATION

We proposed optimizing method for OMG to optimize the constraint on both parameters of the neural networks $\theta_w$, and parameters of the Orthogonal Group Mapping $\theta_{map}$. We use a two-step approach to optimize $argmin_{\theta_{map}, \theta_w} L$ by optimizing $argmin_{\theta_w} L$ and $argmin_{\theta_{map}} L$ iteratively, where $\theta_w$, is the parameter of original network, and $\theta_{map}$ is the parameter of Orthogonal Group Mapping. For the first step, we optimize the weights of the net $\theta_w$ with SGD. For the second step, we optimize the weights of mapping $\theta_{map}$ with Algorithm 1.

**First step**: We use the standard SGD to optimize $argmin_{\theta_w} L$:

$$\theta_w := \theta_w - \eta \nabla(\theta_w) = \theta_w - \eta \sum_i L_i(\theta_w)$$

Where $\eta$ is the step size, and $L_i(\theta_w)$ is the value of the loss function at $i_{th}$ iteration.

**Second step**: We propose Algorithm 1 to optimize $argmin_{\theta_{map}} L$:

---

**Algorithm 1** Optimization Algorithm

---

Initialization
Random Initialize Mapping $\theta_{map}$
Given a batch of training set
**for** *each iteration* **do**
 **for** *each group $g_i$, $i \in [1, K]$* **do**
 Find the max violated unit $n_l$ in $g_i$ that $l \in g_i$ by:
 $l = argmax_l \sum_{m \in g_i} (A_l - A_m)^2$
 Reassign the unit $n_l$ from $g_i$ to new group $g_k$ where:
 $k = argmin_k \sum_{m \in g_k} (A_l - A_m)^2$
 **end**
**end**

---

It is easy to see that, optimizing $argmin_{\theta_{map}}L$ will not change any parameter of the original deep network $\theta_w$. That is to say, given any pre-train network, we could switch the network to an orthogonal grouping network without changing its original parameters.

## 3.2 Dimension Reduction and $k$-Shot Learning

Given an Orthogonal Method of Grouping with corresponding $\theta_w$ and $\theta_{map}$, we could assign $L$ outputs to $K$ groups. The outputs in each group is correlated. we assign an additional weights $w^{add} \in R^K$ and bias $b^{add} \in R^K$ corresponding to each group. We omit the bias for simplicity. For each group $g_i$ it shares the same weight $w_i^{add}$. While doing $k$-shot learning on a few samples, the original weights $\theta_w$ are fixed, and only the $w^{add}$ is update. In another view, we could regard this as an end-to-end dimension reduction method. It could reduce the original output dimension $L$ to arbitrary dimension $K$. Each dimension in the $K$ is largely orthogonal to each other.

## 4 Dataset

The Orthogonal Method of Grouping is evaluated on three standard datasets: ImageNet, MNIST and Office Dataset. ImageNet (Alex et al. (2012)) is the largest publicly available dataset with image category labels. The MNIST (LeCun et al. (1998)) dataset of handwritten digits contains a training set of 60K examples and a test set of 10K examples. The Office (Saenko et al. (2010)) dataset is a collection of images from three distinct domains: Amazon, DSLR, and Webcam. The dataset contains objects of 31 categories commonly spotted in office, such as keyboards, file cabinets, and laptops. Among the 31 categories, there are 16 overlaps with the categories present in the 1000-category ImageNet classification task. This dataset is first used by Judy et al. (2013c) for $k$-shot adaptation task and we follow the same data split of their work.

## 5 Evaluation

We evaluate the OMG from three different tasks: training from scratch, finetuning and $k$-shot learning. We report the performance for training from scratch on MNIST to show that OMG could help to improve the performance by cast orthogonal constraint. Then we smoothly integrated OMG into standard pre-trained networks to show OMG can successfully enhance the orthogonality among groups of parameters on arbitrary neural networks, and results in more discriminative parameters. For $k$-shot learning tasks, we report our $k$-shot learning results on MNIST and Office Datasets (Saenko et al. (2010)). Experiments show our learned compact orthogonal parameters could facilitate learning classifier on limited data.

In the following sections, Experiments for training from scratch are reported in 5.1, Experiments for finetuning are reported in 5.2 and the k-shot learning experiments are reported in Sec. 5.3.

## 5.1 Training From Scratch

We show that our OMG facilitates the performance of neural network during the training. We trained a standard Convolutional Neural Networks (Y. et al. (2003), which reported achieving 1.19% error rates) on MNIST dataset as a baseline. For OMG model, we report the difference of accuracies with different $\alpha$, $\beta$, and group size. The difference of accuracy here denotes the difference between baseline's accuracy and the proposed model's accuracy. For example, if the proposed model has 98% of accuracy and the baseline is 97%, then the difference of accuracy is 1. OMG is used to train every convolutional and fully connected layer in the baseline convolutional network rather than one specific layer.

We train OMG from scratch with the different set of $\alpha$ and $\beta$. We set $\alpha$ or $\beta$ to 0, $1e^{-6}$, $1e^{-5}$, $1e^{-4}$, $1e^{-3}$, $1e^{-2}$, $1e^{-1}$ separately, and keep the other hyperparameter to 0. The group size is set to half of the neural unit's size when evaluating the effectiveness of $\alpha$ and $\beta$. We report the difference of accuracy in Table. 1 and Fig. 4. It is easy to see that when $\alpha$ and $\beta \in (0, 1e^{-3})$, the OMG can boost the performance of the convolutional neural network. We find when the value of hyperparameters is around $1e^{-3}$, the generated gradient from $L$ is about 1 - 5% of the gradient from $L_{sigmoid}$, which is a reasonable ratio (hyperparameter of $L2$ normalization is around 1 - 5%). In Fig. 4, it

shows when the hyperparameters are extremely small, the OMG will not change the performance of the network. When the hyperparameters are too large, the constraint will be too strong to learn the parameters properly. When the hyperparameters are in ideal range, then the OMG is casting the orthogonal constraint on the network and force the network to learn more discriminative parameters. This shows the OMG is working smoothly with normal neural networks and able boosting its performance. For the following experiments, we set the $\alpha$ to $5e-5$ and $\beta$ to $1e-4$ if we do not specific mentioned. Theoretically, the $L_{intra}$ would force the filters in each group to be similar, so intuitively, it would jeopardize the performance of the original network. However, as we find in Fig. 4, $L_{intra}$ can actually boost the performance. This is because practically the $L_{intra}$ is being used as a regularization term.

We report the effect of group size as shown in Fig. 5. In Fig. 5 when the group size is set as the same as the neural unit size, the OMG is not really grouped. But the network still has a better performance since the OMG enforces the network to learn more discriminative parameters. The best performance is achieved when the group size is around half of the neural unit size. If the group size is too small, it will force all the neural units to learn the same parameters, which would jeopardize the performance of the network. When the group is $1/64$ of the neural unit size, it achieves the worst performance. For the following experiments, if we do not mention specifically, we set the group size as half of the neural unit size.

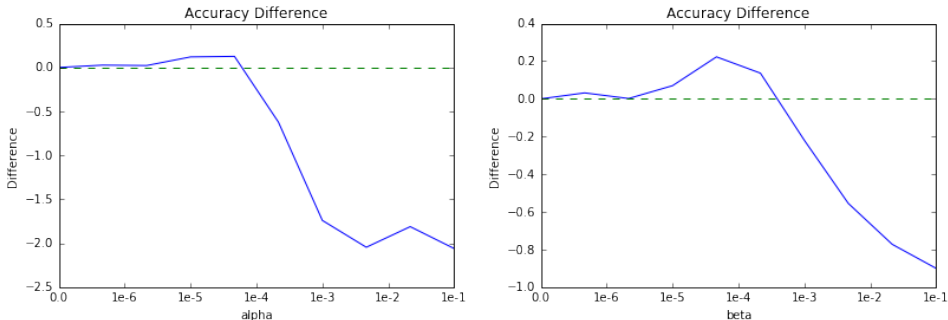

Figure 4: The difference of accuracy between proposed model and baseline as functions of $\alpha$ and $\beta$. The dashed line in each charts is the baseline performance. As we can see from the charts, when alpha and beta are in the range of $[1e^{-6}, 1e^{-4}]$ and $[1e^{-6}, 1e^{-3}]$, the performance of OMG outperforms the baseline.

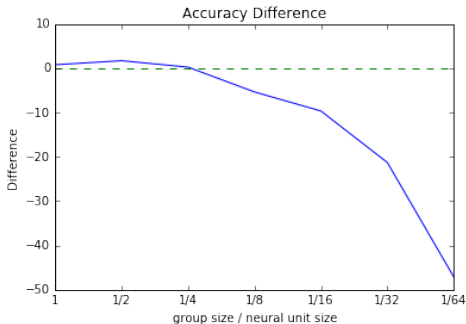

Figure 5: The difference of accuracy between our proposed model and the baseline as functions of a ratio between group size and neural unit size. From the chart, when the number of groups is around [1, 4], the proposed model outperforms the baseline.

Although the OMG can work smoothly with different types of layers as convolutional layer, fully connected layer, and etc, we only visualize the orthogonal groups of first convolutional layer in Fig. 6 since it is the easiest to observe. It is hard to visualize the rest of convolutional layers since the filter size is smaller. And it is also hard to visualize the fully connect layer since we can not observe

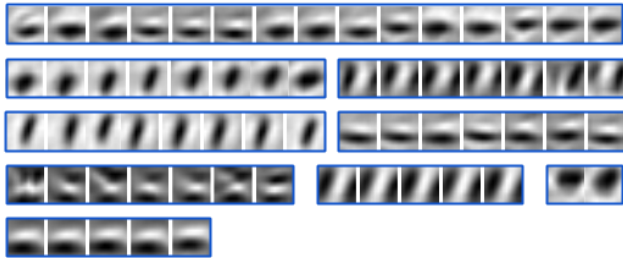

Figure 6: The Visualization of the orthogonal grouped filter maps in the first convolutional layer. The filters in each blue block belong to the same group. It is easy to see that filters in each group share the similar appearance, and the filters from different groups are different in terms of appearance.

clear pattern directly. For a better visualization, we choose the convolutional neural network with $11 \times 11$ kernels. Filters in the same blue bounding boxes belong to the same group. We can see filters in each different groups are highly correlated to each other, and the filters in the different group are visually very different.

| Difference of Accuracy | 0 | $1e^{-6}$ | $1e^{-5}$ | $1e^{-4}$ | $1e^{-3}$ | $1e^{-2}$ | $1e^{-1}$ |
|---|---|---|---|---|---|---|---|
| $\alpha$ | 0 | 0.01 | 0.12 | -0.12 | -1.74 | -1.95 | -2.06 |
| $\beta$ | 0 | 0.01 | 0.07 | 0.23 | -0.22 | -0.68 | -0.9 |

Table 1: The effect of different hyperparameters $\alpha$ and $\beta$

## 5.2 FINETUNING

In this section, we prove that OMG forces deep neural network learn more discriminative features, and the OMG works smoothly with pre-train neural networks. We finetune existing pre-trained networks on ImageNet with the help of OMG. The original network is fully trained on ImageNet dataset, so directly finetune on the same dataset should not change any thing significantly. Even finetune on different datasets, empirically the parameters in the very first layers are not going to change. However, we observe the significant changing after finetuning on the same dataset for only 5 epoches.

We choose to visualize VGG-F (Chatfield et al. (2014)) net since it has the largest kernel size among the VGG zoo ($11 \times 11$ rather than $7 \times 7$). We visualize the grouped filters in the first convolution layer in Fig. 7. It is bacause the same reason, we only visualize the first convoutional layer The kernels in the same vertical line belong to the same group. We have 20 groups and each group has the number of kernels from 1 to 6. It is easy to see that the filters within each group share similar appearance (highly correlated), and the filters from different groups have divergent appearances. For example, the second and the $8_{th} - 13_{th}$, they are all white and black edges with the same direction. the kernels in the $6_{th}$ group are all like gaussian kernels with a little jitter in location.

In order to prove that the OMG can help to learn the more discriminative parameters, we visualize the filters after finetuning in Fig. 8. The filters on the left have strong pattern before finetune, and they do not change much after finetuning. For the filters on the right, the filters do not have the strong patterns, but after finetuning with OMG, the pattern of the filters become more distinct with strongly colorful textures. Our interpretation is, our OMG assigns additional orthogonal constraint to all the filters, force them to learn more discriminative parameters. The filters with strong patterns are not changed too much because they are originally orthogonal to other filters. But for the filters without strong patterns, the orthogonal constraint are highly effective. As a result, the OMG helps the VGG network pretrained on ImageNet to learn the new discriminative paramters during finetuning on the same dataset.

## 5.3 $K$-SHOT LEARNING

The performance for $k$-shot learning is evaluated on two different task: MNIST k-shot learning and Office k-shot learning tasks.

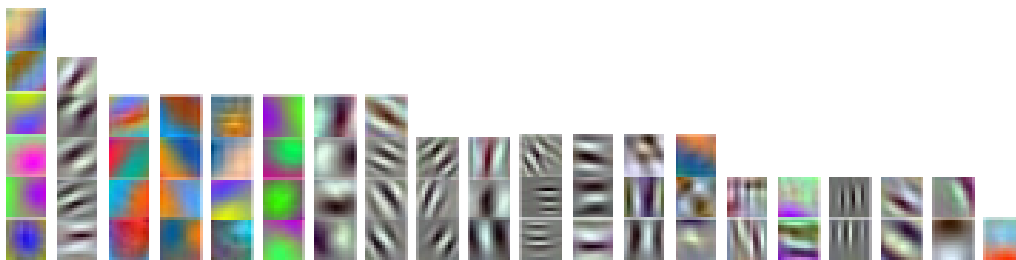

Figure 7: Visualization of the orthogonal grouped filter maps in the first convolutional layer. Filters in each row belong to the same group. It is easy to see that filters in each group share the similar appearance, and the filters from different groups are different in terms of appearance.

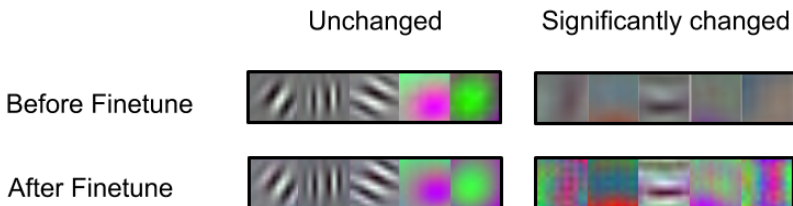

Figure 8: Visualization of the filter maps from the first convolutional layer. 10 filters are selected from the original filter map to show what have the OMG learned. The first column shows the filters from the original VGG-F. The second column is the corresponding filters after finetune with OMG. The filters on the left originally have the strong patterns, and they are mainly unchanged after finetuning. The right ones do not have strong pattern originally, and the pattern becomes more distinct after finetune with OMG. All the filters are normalized to the same magnitude before visualization.

### 5.3.1 $K$-SHOT ON MNIST

We perform $k$-shot learning on MNIST dataset. The performances are evaluated on a 10-way classification where each class is provided with 1, 5 training examples. For the MNIST one shot learning task, we split the data to pre-knowledge set, and one shot learning set with a ratio of 1:9. The models we compared with are k-Nearest Neighbors(K-NN), Support Vector Machines(SVM), Traditional Convolution Neural Networks, and Deep Boltzmann Machines (DBM), and compositional patch model (CPM). The CPM (Alex & Yuille (2015)) is a model designed for learning a compact dictionary of image patches representing meaningful components of an object. The performances were evaluated on a 10-way classification where each class is provided with 1 and 5 training examples to show the growth in accuracy. For a given run, each model is given a set of hand-written digits picked at random from each class from the one-shot learning set. For CNN, we used a structure with four convolutional layers and two pooling layers. For DBM, it contains two hidden layers with 1000 units each. For the OMG model, it learns the grouping information on pre-knowledge set without using ground truth. Then it is trained on the samples from one shot learning set, where the group sizes are set by grid search. The performance is reported in Table. 2. The performance of OMG is better than the baselines, especially better than the previous CNN method. It is because the huge parameter space of original CNN is hard to optimized by only a few samples. But with the help of OMG, the parameter space is significant reduced. The OMG can learn the group of filters representing the shared common parts as each part holds immense amounts of information on how a visual concept is constructed. And using these patches as features to learn a better classifier with limited data.

| Methods | Sample n = 1 | Sample n = 5 |
|---------|--------------|--------------|
| DBM | 24.37 | 41.76 |
| CNN | 28.01 | 39.8 |
| K-NN | 42.08 | 64.26 |
| SVM | 2.78 | 10.08 |
| CPM | 68.86 | 83.79 |
| OMG | 70.17 | 84.35 |

Table 2: Comparison of accuracy with other models on $k$-shot learning tasks. The proposed OMG achieves better performance on both one-shot learning and 5-shot learning case.

### 5.3.2 $K$-SHOT ON OFFICE DATASET

In this section, we conduct the experiment on k-Shot learning task on Amazon Office Dataset (Saenko et al. (2010)). We followed the work described in Judy et al. (2013c), conduct k-shot learning on the 16 categories of Office dataset (approximately 1,200 examples per category or 20K images total). We evaluated our method across 20 random train/ test splits, and each test split has 160 examples. Then the averages error are reported. For each random train/ test split we choose one example for training and 10 other examples for testing. Following the previous work, we use pre-trained DECAF from ImageNet. The model is additionally trained with Orthogonal Method of Grouping on the last 3 convolutional and fully connect layers, where the group sizes are set by grid search. The group size of each layer is always around half of the neural unit size of the layer. Then we perform $k$-shot learning on the Office dataset with the reduced dimension. The accuracy is reported in Table. 3. The models we compared with are SVM, PMG, Daume III and Late fusion. Late fusion (Judy et al. (2013b)) is a simple approach to independently train a source and target classifier and combine the scores of the two to create a final scoring function.

| Methods | Accuracy |
|---------|----------|
| SVM | 62.91 |
| Late Fusion (Max) | 59.59 |
| Late Fusion (Lin. Int. Avg) | 60.64 |
| Late Fusion (Lin. Int. Avg) | 71.10 |
| PMT | 64.84 |
| Daume III | 68.89 |
| OMG | 71.97 |

Table 3: One shot learning result on Office dataset, the numbers of baseline are borrow from Judy et al. (2013b).

We achieve the best performance comparing to previous state-of-the-art method Judy et al. (2013c). With the help of OMG, the dimension reduced network achieve a better performance when training on limited data. The main reason is because, the original feature space on DeCaf is large and redundant. By grouping the feature space, the dimension is largely reduced and the network is still representative enough for the Office Dataset. Then a better performance is achieved on Amazon one shot learning task with OMG.

## 6 CONCLUSION

We proposed a generalizable $k$-shot learning framework that can be easily integrated into any existing deep net architectures. By grouping parameters together and forcing orthogonality among groups, the method is able to reduce parameter space dimensionality for avoiding overfitting. Experiments on $k$-shot learning tasks have proven that OMG is able to have a good performance on $k$-shot class and easily to be adopted to the current existing deep net framework like VGG, ResNet and so on.

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
