# Peer review of "OMG: Orthogonal Method of Grouping With Application of K-Shot Learning"

_ICLR 2017 — rejected_

[Official Review · AnonReviewer1 · rating 4 · confidence 5 · 15 Dec 2016]
**Dimensionality Reduction Approach.**

The authors of this work propose a learnable approach to reducing the dimensionality of learned filters in deep neural networks. This is an interesting approach, but the presented work looks a bit raw.

1. There are many typos in this manuscript. 
2. The experimental results are rather weak and don't show much improvement in accuracy. Instead the authors could position this work as a compression mechanism and would have to compare to low rank approximation of filters for DNNs. Yet this is not done. 
3. Aside from compression, OMG can be viewed as a form of regularization to reduce the unnecessary capacity of the network to improve generalization. Again, this is not addressed in enough detail.
4. If the authors care to compare their approach to other 1-shot learning methods, then they would have to evaluate their approach with siamese and triplet learning networks. This isn't done.

[Official Review · AnonReviewer2 · rating 4 · confidence 4 · 17 Dec 2016 (modified: 23 Jan 2017)]

This paper proposes a regularization technique for k-shot learning based on orthogonal grouping of units in a neural network. The units within a group are forced to be maximally similar, at the same time the units from different groups are encouraged to be orthogonal. While I like the motivation of the approach, the empirical analysis provided in the paper doesn’t look particularly convincing.

My main concerns are the following:

1. The method is sensitive to the values of alpha and beta and a poor choice of those hyperparameters can lead to a quite drastic drop in performance comparing the minor gains one gets when alpha and beta are set properly.

2. It seems strange that the best performance is obtained when the group's size ratio is 0.5. From the figures in the paper, it follows that usually, one has more “orthogonal” groups in a filter bank. I have an impression that the empirical evidence doesn’t align well with the motivation of the proposed approach.

3. The paper contains a significant amount of typos and incorrectly formatted references. There are also several places in the manuscript that I found hard to understand due to unusual phrasing.

I would like to thank the authors for answering/addressing my pre-review questions. I would be grateful if the authors could provide more clarifications of the following:

1. Question 2: I’m not sure if modifying \theta_{map} alone would result in any learning at all. Do I understand correctly that \theta_{map} is only used to define groups? If so, then I don’t see how the proposed method can be used in the purely unsupervised regime.

2. Question 3: I was not referring to the fixed clustering based on the filter of the pre-trained network. One can perform that clustering at every step of the k-shot learning process. I’m not sure I understand why the authors visualize grouping of _filters_ while in the actual algorithm they group _activations_. 

Overall, the paper is quite interesting but needs a stronger empirical justification of the approach as well as a better presentation of the material.

[Official Review · AnonReviewer3 · rating 4 · confidence 4 · 21 Dec 2016]

This paper proposes a k-shot learning framework that can be used on existing pre-trained networks by grouping filters that produce similar activations. The grouped filters are learned together to address overfitting when only few training samples are available. 

The idea of the paper is interesting there are some encouraging results, but the current version doesn't seem ready for publication:

Performance:
The method should be compared with other state-of-the-art k-shot learning methods (e.g., Matching Networks by Vinyals et al., 2016). It's not clear how this method compares against them.

Missing explanation:
Experimental setting for k-shot learning should be more detailed.

Measure:
Accuracy difference does not look like a good idea for comparing the baseline method and the proposed one. Just raw accuracies would be fine. 

Many grammatical errors and inappropriate formatting of citations, such as:
M. et al. (2011)
ImageNet (Alex et al. (2012))
Judy et al. (2013): this reference appears three times in the reference section.

[Final Decision · Program Chairs · 06 Feb 2017]
**ICLR committee final decision**

All three reviewers point to significant deficiencies. No response or engagement from the authors (for the reviews). I see no basis for supporting this paper.